# Revisiting Alpha-Synuclein Pathways to Inflammation

**DOI:** 10.3390/ijms24087137

**Published:** 2023-04-12

**Authors:** Patrícia Lyra, Vanessa Machado, Silvia Rota, Kallol Ray Chaudhuri, João Botelho, José João Mendes

**Affiliations:** 1Egas Moniz Center for Interdisciplinary Research (CiiEM), Egas Moniz School of Health and Science, Caparica, 2829-511 Almada, Portugal; vmachado@egasmoniz.edu.pt (V.M.); jbotelho@egasmoniz.edu.pt (J.B.); jmendes@egasmoniz.edu.pt (J.J.M.); 2Evidence-Based Hub, CiiEM, Egas Moniz—Cooperativa de Ensino Superior, Caparica, 2829-511 Almada, Portugal; 3Department of Basic & Clinical Neuroscience, Institute of Psychiatry, Psychology & Neuroscience, King’s College London, London WC2R 2LS, UK; silvia.rota@kcl.ac.uk (S.R.); ray.chaudhuri@kcl.ac.uk (K.R.C.); 4Parkinson’s Foundation Center of Excellence, King’s College Hospital, London SE5 9RS, UK

**Keywords:** alpha-synuclein, inflammation, neuroinflammation, synucleinopathies, Parkinson’s disease

## Abstract

Alpha-synuclein (α-Syn) is a short presynaptic protein with an active role on synaptic vesicle traffic and the neurotransmitter release and reuptake cycle. The α-Syn pathology intertwines with the formation of Lewy Bodies (multiprotein intraneuronal aggregations), which, combined with inflammatory events, define various α-synucleinopathies, such as Parkinson’s Disease (PD). In this review, we summarize the current knowledge on α-Syn mechanistic pathways to inflammation, as well as the eventual role of microbial dysbiosis on α-Syn. Furthermore, we explore the possible influence of inflammatory mitigation on α-Syn. In conclusion, and given the rising burden of neurodegenerative disorders, it is pressing to clarify the pathophysiological processes underlying α-synucleinopathies, in order to consider the mitigation of existing low-grade chronic inflammatory states as a potential pathway toward the management and prevention of such conditions, with the aim of starting to search for concrete clinical recommendations in this particular population.

## 1. Introduction

Alpha-synuclein (α-Syn) is a monomer that is expressed at different levels, as it seems to be present not only in nerves—at the presynaptic nerve terminals of both the Central Nervous System (CNS) and the Peripheral Nervous System (PNS) [1]—but also at the erythrocytes and immune cells level [2]. Intracellularly, α-Syn can be found in membranes and in the cytoplasm, as well as in several organelles, such as the nucleus, mitochondria, endoplasmic reticulum, golgi apparatus, and the endolysosomal system, even though its role on these organelles still remains unclear [3,4,5,6,7,8,9,10,11]. This small soluble monomer consists of 140 amino acids encoded by the α-Syn gene on the long arm of human chromosome 4 [12]. Under physiological conditions, α-Syn is thought to be involved in the regulation of synaptic vesicle traffic, particularly neurotransmitter release and reuptake [13,14]. Specifically, α-Syn acts in vesicle priming, fusion, and dilation of exocytotic fusion pores [15]—as illustrated in Figure 1—increasing the local release of calcium (Ca^2+^), which is critical for ATP-induced exocytosis [16] and modulates the dopamine transporter (DAT1), regulating dopamine neurotransmission [17]. Furthermore, α-Syn’s multimeric membrane-bound state presents molecular chaperone activity, essential to sustaining normal SNARE-complex formation during aging (demonstrated in Figure 1), as it promotes the folding of synaptic fusion components called Soluble NSF Attachment Protein REceptors (SNAREs) at the presynaptic plasma membrane in conjunction with cysteine string protein-alpha/DnaJ Heat Shock Protein Family (Hsp40) Member C5 (DNAJC5) [18]. Furthermore, in terms of its immunomodulatory function, α-Syn seems to be involved in the development of B lymphocytes and in the regulation of T cells [19], playing different roles in either promoting disease pathogeny (as is the case of PD) and in protecting against proinflammatory responses or infections [2,20].

There is a dynamic balance between the different three-dimensional forms that α-Syn can undertake, varying from the monomeric and oligomeric forms to the formation of fibrils, which does not easily occur in a homeostatic environment [21]. Evidence shows, however, that under pathological conditions (in which an environmental trigger such as the settlement of a virus or bacterial infections have been hypothesized), α-Syn monomers can undergo post-translational modifications that cause overexpression and an increase in intracellular levels, fostering its aggregation and development of toxic α-Syn oligomers and protofibrils [13,22,23,24], which subsequently spread throughout the CNS.

Indeed, the gastrointestinal tract or even exposed olfactory neurons have been suggested as possible entry routes for these pathogens [20]. The α-Syn toxic species act through cell-autonomous mechanisms in addition to non-cell-autonomous ones [25]. The former may disrupt organelle activity (by increasing oxidative stress or by impairing the ubiquitin-proteasome machinery and mitochondrial function). This organelle disruption may be responsible for neuronal cytotoxicity, while the latter may be responsible for inducing synaptotoxicity and affecting the distribution and activation of synaptic proteins, preventing neurotransmitter exocytosis and neuronal synaptic communication [22,26] and possibly the recruitment and activation of glial cells (resident immunocompetent cells of the CNS) [27]. This could ultimately lead to the neurodegeneration cascade [28,29,30]. In addition, the perforation of the neuronal plasma membrane is enabled by extracellular α-Syn oligomers, increasing its conductance and the influx of Ca^2+^ ions and glucose, partially supporting the synaptotoxicity observed in α-synucleinopathies [31]. Figure 2 represents the physiological and pathological paths of α-Syn.

The transmission of α-Syn pathology spans throughout various brain regions, especially dopaminergic neurons in the substantia nigra pars compacta (the core site of neurodegeneration in PD), although the full extent of its impact remains largely unknown [32]. Furthermore, α-Syn can be detected as a biomarker in PD patients’ cerebrospinal fluid (CSF), saliva, serum, urine, and gastrointestinal tract [33,34,35,36]. Several studies have also attempted to measure α-Syn in blood cells and plasma, but so far, the results have been inconsistent [37].

Indeed, PD is considered to be the most common synucleinopathy [22], and represents the fastest growing and most significant medical and social burden of our time [38]. The α-Syn gene was the first gene to be associated with PD, and the knowledge that a single missense/point mutation or duplications of the gene cause Parkinsonism, especially the autosomal dominant form, supports the strong association between α-Syn and PD [12,39,40]. Nevertheless, it is noteworthy that non-genetic (environmental and idiopathic) factors appear to be responsible for the vast majority of PD cases [41] and have led to the consideration of the CRISPR/Cas9-mediated technology far beyond in this regard [42,43]. In terms of its pathological features, PD is characterized by the loss of dopaminergic neurons in the midbrain region of the substantia nigra pars compacta, resulting in a marked decrease in dopamine levels in the synaptic terminals, as well as the presence of intraneuronal dense inclusions of Lewy bodies (aggregations of misfolded α-syn, ubiquitin, complement proteins, and cytoplasmic structural proteins) in different regions of the brain and body (substantia nigra pars compacta, cerebral cortex, dorsal nucleus of the vagus nerve, sympathetic ganglia, and myenteric plexus of the gut) [22,44,45,46]. Overall, PD is a highly systemic and multifactorial neurodegenerative disease whose motor and non-motor symptoms (NMSs) have a major impact on patients’ quality of life. [26,45,47,48]. Even though PD is only diagnosed upon the detection of its progressive cardinal motor symptoms (resting tremor, muscular rigidity, and bradykinesia), often together with a variety of NMSs [49,50], a prodromal phase is believed to occur up to 30 years prior to clinical diagnosis, which includes signs such as hyposmia (olfactory dysfunction), sleep abnormalities (Rapid eye movement (REM)-sleep behavior disorder), cardiac sympathetic denervation, constipation, depression, and pain [41,51,52]. During this early prodromal phase of PD, synaptic alterations occur prior to neuronal death, namely the accumulation of toxic α-Syn in the presynaptic terminals, affecting neurotransmitter release [22].

In fact, PD presents a multifactorial etiology that remains unclear thus far, even though it seems to be dependent both on environmental and genetic factors, while encompassing several biological mechanisms such as α-Syn pathology, mitochondrial dysfunction, oxidative stress, synaptic plasticity, neuroinflammation, chronic systemic inflammation (translated in the dysregulation of circulating inflammatory cytokines), and even gut and periodontal dysbiosis [53,54]. Even though α-Syn dysfunction is a prominent key component of PD and other α-synucleinopathies, such as dementia with Lewy bodies (DLB) and multiple system atrophy (MSA), the reason it accumulates is still an active area of research [45]. To that end, the intent of this review is to revisit a-Syn pathways to inflammation, aiming to clarify and update the available evidence to date, as well as to promote further research in the hopes of ultimately establishing future clinical recommendations.

## 2. α-Syn and Inflammation

Recent evidence from both in vitro and in vivo models report on the major role neuroinflammation plays on the pathophysiology of neurodegenerative disorders, including PD, as it is pivotal factor between PD’s genetic predisposition and environmental exposures [55,56,57,58,59]. In fact, the use of non-steroidal anti-inflammatory drugs appears to reduce the risk of developing PD [60]. It is known that α-Syn accumulation stands as a key component of these neuronal inflammatory patterns, in particular, by modulating microglial function and upregulating the inflammatory cascade [61]. In detail, toxic α-Syn species that are translocated intracellularly can trigger microglia hyperactivity, activate astrocytes, increase the gene expression of proinflammatory factors, and summon peripheral immune cells to the surroundings of the pre-apoptotic and apoptotic dopaminergic neurons on the CNS, all of which might induce neuronal dysfunction [22,62,63].

Therefore, alongside peripheral lymphocyte infiltration, there is an in situ increase in inflammatory mediators such as interleukin (IL)-1β and inflammasome cytosolic nod-like receptor protein 3 (NLRP3) in PD patients’ CSF and brains [64,65]. The chronic overexpression of IL-1β at the substantia nigra pars compacta level of mice culminates in dopaminergic degeneration through glial activation and motor deficits [27]. The same inflammatory burden was systematically reported in animal studies, with reports of NLRP3 activation in the serum of PD mice, and in human trials with elevated peripheral blood levels of NLRP3, interleukin (IL)-6, tumor necrosis factor-alpha (TNF-α), IL-1β, IL-2, IL-10, C-Reactive Protein (CRP), and Regulated upon Activation, Normal T Cell Expressed and Presumably Secreted (RANTES) in PD patients [53,66,67]. This evidence points out NLRP3 and IL-1β as major inflammatory candidates with serological diagnostic potential in PD [66,68]. On the one hand, NLRP3 can be activated by toxic α-Syn peripherally and released in human monocytes and microglial cells [67,69,70], amplifying the inflammatory response [71], while also secreting IL-1β from peripheral blood mononuclear cells and microglia [67,69,70]. On the other hand, a rare NLRP3 polymorphism is associated with decreased PD risk [72]. In sum, inflammogens IL-1β and NLRP3 might be involved in PD neurodegeneration, onset, and progression [72,73]; however, further research on the α-syn–inflammasome relationship is warranted.

Furthermore, the IL-6 inflammogen is known for its role in neuropathology and is thought to present both pro-inflammatory and anti-inflammatory function in PD, with 2.3-fold higher concentration levels at the periphery in PD patients when compared to controls [74].

## 3. Microbial Dysbiosis and α-Syn

The microbiota—which consists of thousands of bacterial, viral, and fungal species that inhabit different parts of the human body—plays a critical role in human health, not only through its barrier function against pathogens, but also through its regulatory role of the immune system as well as its impact on other important functions, such as the regulation of movement [75]. The human gut microbiota in particular has been the focus of intense research. This microbiota is shaped by lifetime determinants (such as diet, disease history, age, or genetic heritance) [76,77] and produces a variety of molecules, some of which can enter the bloodstream and affect overall systemic health [78]. In a recent cross-sectional study, gut microbiota and plasma metabolites of 8583 participants of the population-based Swedish CArdioPulmonary bioImage study, with ages comprised between 50 to 64 years, were studied [79]. Using metagenomics and ultra-high-performance liquid chromatography combined with mass spectrometry, Dekkers et al. [79] found that 58% of the individual variance of plasma metabolites were explained by a specific gut microbiome. Similar results were found by Diener et al. [80], by analyzing 930 blood metabolites against genetics and gut microbiome variation in 1569 individuals. Overall, 69% of the found associations were the result of sole microbiome interactions, 15% due to genetic reasons, and 16% a hybrid genomic–microbiome interaction [80].

Therefore, disruptions on microbiota present the potential to lead to broad immune dysfunction with possible neuronal consequences [45]. Thus, a dysbiotic phenomenon of the commensal microflora may be a precursor of diseases, especially inflammatory ones, as is the case of irritable bowel syndrome, periodontitis, liver disease, rheumatoid arthritis, obesity, diabetes, or even neurological disorders, as is the case of depression, anxiety, and PD [45,81,82].

The endotoxin lipopolysaccharide (LPS), being the Gram-negative bacteria’s main membrane component that hampers phagocytosis by immune system cells, stands out as one of the major bacterial defense mechanisms [45]. Recent evidence based on α-Syn-induced mice exposed to LPS (intraperitoneally injected) reported cognitive deficits and enhanced dopaminergic neuronal loss, as α-Syn adopted its fibrillar form, which suggests that LPS-induced neuroinflammation and the PD-related genetic background interact synergically [27,83]. Therefore, bacterial exposure may be a driving force in α-synucleinopathies [84,85]. LPS also induces the expression of chemokines [86,87,88,89].

### 3.1. Gut Microbiota and α-Syn

Recent lines of research have been exploring the potential mechanisms by which changes in the gut microbiota and their products (such as LPS and intestinal-mucosa-derived inflammatory factors) might contribute to the misfolding and abnormal aggregation of α-Syn in the enteric nervous system (ENS), and upon transportation via projections of the vagus nerve and autonomic enteric fibers, in the CNS [43,77,78]. The transport of toxic α-Syn species through the microbiota–gut–brain axis ultimately results in the loss of dopaminergic neurons and causes a microglial inflammatory response that is in the pathogenesis of α-synucleinopathies [83]. In fact, PD patients appear to present differences regarding the level of certain species of gut bacteria when compared to healthy counterparts [84,85,86,87]. Therefore, it is crucial to highlight the microbiota–gut–brain axis, as it represents a complex and interdependent network between the ENS, the gut microbiota, the immune system, and the brain and provides key insights into how intestinal alterations might affect distant organs, such as the brain [88]. In fact, the gastrointestinal tract’s communication network with the CNS includes pathways such as the systemic circulation of hormones, inflammatory cytokines, and microbial products, as well as the autonomic nervous system through the vagus nerve [52]. Interestingly enough, of the non-motor features that comprise the prodromal phase of PD—which include gastrointestinal, olfactory dysfunction, autonomic dysregulation, fatigue, sleep disorders, and mood disturbances—the early constipation and gastrointestinal inflammation support the involvement of the microbiota–gut–brain axis [87,88,89,90,91]. However, even though the gut microbiota’s role in neurodegeneration has started to be explored, research on the involvement of oral microbiota on such mechanisms is still due [87,92].

In a recently conducted study, Shi et al. [90] explored whether mucosal microbiota in PD patients would correlate with changes in intestinal mucosal a-Syn. To this end, nineteen PD patients were compared to healthy counterparts for duodenal and sigmoid mucosal samples with next-generation metagenomic sequencing. Overall, the results showed that oligomer a-Syn in the sigmoid mucosa is transferred from the epithelial intestinal wall to the cytoplasm, acinar lumen, and stroma. Furthermore, the intestinal mucosal microbiota composition changed with the increase in the relative abundance of pro-inflammation inducing bacteria in the duodenal mucosa [90]. Beyond the unequivocal potential for diagnosis of PD using such samples, these results show that intestinal microbiota may have a role in the levels of a-Syn at the intestinal mucosa.

### 3.2. Oral Microbiota and α-Syn

There is a clear clinical association between periodontitis and PD that has been centered on both the progressive installment of motor disturbances and cognitive decline, which implicate on the patient’s self-care ability and compromises oral hygiene, as well as fewer dental attendances [54], which ultimately precipitate oral diseases [91,92,93]. However, the extent of the existing evidence regarding a concrete α-Syn or other crosstalk biomarkers linking periodontitis and PD is sparce and still relies on a bioinformatic genomic analysis [94,95].

Periodontitis knowingly disturbs systemic health, either through its inflammatory burden or bacterial blood dissemination [96,97,98,99]. In particular, an association between periodontal inflammation and neurodegenerative conditions has been reported in studies regarding cognitive function [100,101], dementia [102], and very recently PD [103]. The hypothesis in which an active periodontal infection promotes the secretion of pro-inflammatory cytokines such as interleukin (IL)-1, IL-6, TNF-α, and reactive oxygen species (ROS) and might increase the risk of PD has been proposed [104], alongside the fact that periodontal infection constitutes a new entryway for bacterial translocation in PD [54]. In addition, not only were *Porphyromonas gingivalis* and its toxic proteases (gingipains) identified in the brain of Alzheimer’s disease (AD) patients [55], but an increase in the deposition of beta amyloid has also been reported in the brain of periodontitis-induced mice models for AD [105]. The presence of these key periodontal pathogens in distant tissues, and their association with inflammation, may suggest that the migration of these microorganisms might cause local inflammatory reactions, and are often related to pathological mechanisms of major neurological diseases.

We have recently demonstrated a higher prevalence of periodontitis in PD patients [103] and possible systemic repercussions with the elevation of circulating white blood cell counts [106] and c-reactive protein [107]. In fact, systemic inflammation caused by periodontitis has been hypothesized to develop chronic neuroinflammation and ultimately interfere with PD pathogenesis [104]. The chronic pattern of periodontitis may be responsible for sustained local and systemic inflammatory states with unknown consequences in PD, perhaps suggesting the involvement of a microbiota–mouth–brain axis, and these results may pinpoint mechanistic clues for future research on the biological mechanisms and paths. In addition, immunologically different traits and patterns mediated by oral microbiota might interfere with the pro-inflammatory state in PD, with the individual as well as horizontal and vertical inheritance having a conceivable role.

The first study to investigate the oral microbiota in PD was based on the hypothesis that this disease is often characterized by neuropathological changes in olfactory and gastrointestinal tissues. Therefore, Pereira et al. compared oral and nasal samples from PD patients with controls [108]. Gene sequencing data revealed a different oral and nasal microbiota in the abundance of individual bacterial taxa, but without significant differences in PD patients. Despite the lack of apparent differences, the authors outlined the potential importance of tracking gut microbiota for potential clinical purposes. Later, Rozas et al. [109] conducted a similar cross-sectional case–control study, examining the oral microbiota from hard and soft tissues of PD patients (and matched healthy controls). The bacterial identification results showed significant differences in soft tissue diversity with a higher abundance of opportunistic oral pathogens in PD. When potential confounders were examined, the presence of dysphagia, drooling, and salivary pH emerged as the most influential. Taken together, these findings revealed novel microbiota differences and clinical signs that could explain them.

Years later, and with new data pointing to the importance of diet in the pathophysiology of PD, Zapała et al. [110] compared the dietary preferences and oral microbiota profile of PD patients with healthy matched controls. Gene sequencing showed that the oral microbiota in PD differed from the controls, with a lower abundance of *Proteobacteria*, *Pastescibacteria*, and *Tenercutes*. On the other hand, a high relative abundance of *Prevotella*, *Streptococcus*, and *Lactobaccillus* was found. In addition, dietary patterns showed correlations with microbial taxa, suggesting a possible role of diet in the oral bacterial profile of PD patients.

In addition, a recent exploratory study examined the composition of oral microbiota (saliva and subgingival samples) and the level of oral inflammation (by periodontal and dental examination and quantification of gingival crevicular fluid levels of IL-1β, IL-6, IL-1 receptor antagonist, interferon-γ, and TNF-α) [111]. Both in saliva and subgingival plaque, the composition of plaque showed different patterns between PD and control subjects [111]. There was a higher abundance of *Streptococcus mutans*, *Kingella oralis*, *Actinomyces AFQC_s*, *Veillonella AFUJ_s*, *Scardovia*, *Lactobacillaceae*, *Negativicutes,* and *Firmicutes*. On the contrary, there was less abundance of *Treponema KE332528_s*, *Lachnospiraceae AM420052_s*, and *phylum SR1* [111]. Even though there were no differences between dental and periodontal statuses among PD and control patients, there was a higher level of IL-1β and IL-1 receptor antagonists in the gingival crevicular fluid of people with PD, showing a different inflammatory pattern on the periodontal apparatus.

The apparent specific oral microbiota profiling in PD raised other questions of major importance regarding the pathophysiological features of this movement disorder. Particularly, Zheng et al. [112] collected oral mucosa samples using a cytological brush from people with PD and age-matched controls. Immunofluorescence analysis revealed increased α-Syn, pS129, and oligomeric α-Syn levels in oral mucosa cells of PD patients. While α-Syn species were distributed intracellularly, pS129 was mainly located in the cytoplasm, and oligomeric α-Syn in the nucleus and perinuclear cytoplasm. In addition, the oral mucosa α-Syn and oligomeric α-Syn levels of participants with PD significantly correlated with the clinical staging of PD, assessed with the Hoehn–Yahr scales.

At that point, several studies were able to identify a distinct gut microbial composition in PD. Jo et al. [113] furthered the research on the microbiome by studying the functional alteration of the microbiome in PD. The taxonomic oral and gut microbiome profile significantly differed between PD patients and healthy controls, with a higher abundance of *Lactobacillus* and opportunistic pathogens [113]. Functional analysis revealed a down-regulation of microbial glutamate and arginine biosynthesis gene markers and an up-regulation of antibiotic resistance gene markers in PD patients compared to healthy controls [113].

At this stage, research has identified changes in the oral microbiota of PD patients, with reductions in some bacterial species and increases in others, but without consistent patterns that may be explained by multiple factors. To this date, we expect the number of studies investigating this particular issue to increase. This body of knowledge may contribute to new frontiers in our understanding of how the oral microbiota, which is an integral part of the gut microbiota, plays a role in PD. On the one hand, these microbial changes can potentially serve as a non-invasive diagnostic tool for PD. By analyzing the genetic material of the oral microbiota, researchers can identify specific bacterial species or gene pathways associated with PD. This information could be used to develop diagnostic tests that can detect the disease earlier and more accurately than current methods. On the other hand, gene sequencing of the oral microbiota could also lead to the development of new treatments for PD. Researchers could identify specific bacterial species or gene pathways that contribute to the disease and target them with probiotics or other therapies. These treatments could potentially slow or even halt the progression of PD. While this is still highly speculative, several lines of evidence point to the key role of the gut microbiota in the CNS, the so-called gut–brain axis, and this should be a central focus of research in the coming years.

## 4. Inflammatory Mitigation and α-Syn

Therefore, and in alignment with recent evidence in which there is a bidirectional communication system amongst the microbiota, the immune, and the nervous systems in PD, the possibility that disease settlement and progression could be stopped indirectly before reaching the brain is revolutionary [45]. In fact, controlling overall cytokine levels significantly improves PD’s motor function [114]. In an animal study using a PD mice model, Manocha et al. explored how neuroinflammation in PD could impact cytokine changes, neuron loss, gliosis, and behavioral dysfunction [114], and then mice had their immune response modulated by a calcineurin/NFAT inhibitor. This evokes the urgency for patients’ global health evaluation, correct clinical diagnosis, and redirection to specialists in order to manage these chronic inflammatory pathologies with the further goal of preventing α-synucleinopathies such as PD and other neurodegenerative diseases [27].

It appears to be crucial to address gut inflammation, particularly in cases of bowel inflammation or periodontal infection. Recent evidence indicates that having poor periodontal health is linked to a higher likelihood of developing PD [104]. Consequently, it is not only essential to encourage patients to improve their oral hygiene practices, but it is also vital to provide them with specialized and ongoing medical care, such as dental scaling, which is the current gold-standard of the non-surgical treatment for periodontitis [115]. By doing so, it may be possible to achieve a protective effect against PD.

Moreover, there seems to be evidence to suggest that various treatments aimed at modulating the gut microbiota can have beneficial effects on PD. For example, fecal microbiota transplantation (FMT) has been shown to have therapeutic potential by restoring the gut microbiota of PD patients and improving their clinical motor and non-motor symptoms, including gastrointestinal symptoms such as constipation [116]. Similarly, pre-, post-, and probiotic therapies have been found to not only affect the clinical scores and some metabolic parameters (such as hs-CRP and insulin metabolism) in human studies, but also to significantly reduce motor impairments related to gait pattern, overall balance, and coordination of movement in a PD mouse model [117,118,119,120,121,122,123]. Additionally, therapeutic interventions in which the administration of antibiotics is involved have been found to present regulatory effects on dopaminergic neurotoxicity in the brain [124,125,126,127]. Furthermore, vagotomy procedures have been associated with a decrease in the risk of developing PD, while also potentially providing neuroprotective effects against PD pathology, which warrants further exploration [128,129].

## 5. Conclusions

To this date, neurodegenerative conditions (such as α-synucleinopathies and PD in particular) are the leading cause of life-limiting disability worldwide. The effects of the microbiota on the misfolding mechanisms of α-Syn in the ENS, the CNS, and in inducing chronic inflammatory states seem to play a key role in the pathogenesis of α-synucleinopathies, even though they are still unclear thus far. In addition, the role of the oral microbiota—which is an integral part of the gastrointestinal tract—as well as its dysbiosis, also needs to be considered in the establishment of the microbiota–gut–brain axis. Therefore, it is urgent to clarify the existing evidence on α-Syn pathways to inflammation, in the hopes of contributing to the establishment of clear pathological processes, as well as to consider the mitigation of these low-grade chronic inflammatory states, as a potential pathway toward the management and prevention of such conditions. Ultimately, the goal is to establish future clinical and robust recommendations with significant impact in this current aging population.

## Figures and Tables

**Figure 1 ijms-24-07137-f001:**
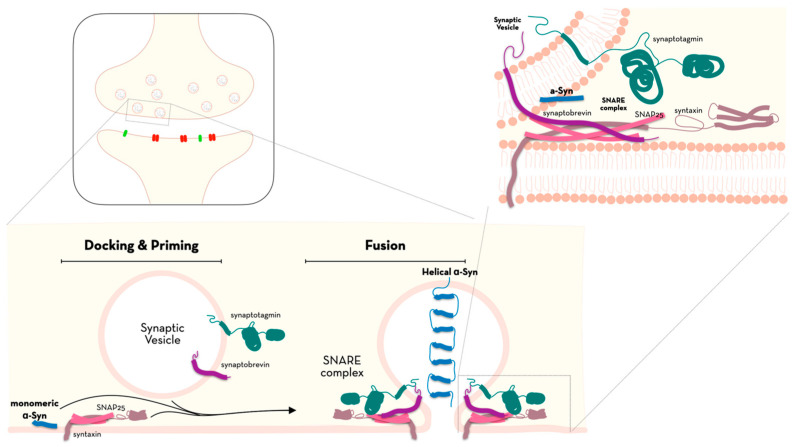
α-Syn’s role in vesicle priming, fusion, and dilation of exocytotic fusion pores and in the promotion of SNARE-complex assembly.

**Figure 2 ijms-24-07137-f002:**
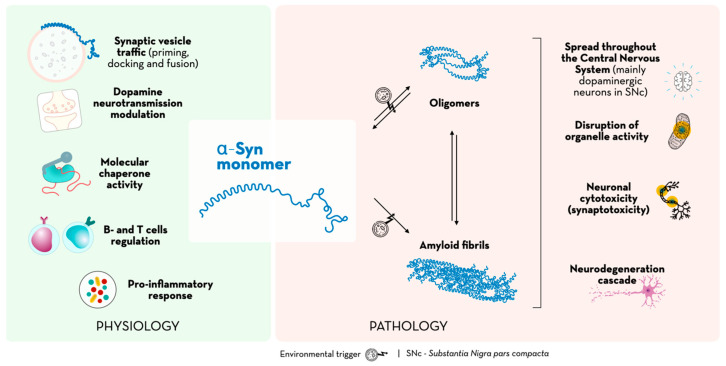
α-Syn’s physiological and pathological paths.

## Data Availability

Not applicable.

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
