# Peer review of "Revisiting Alpha-Synuclein Pathways to Inflammation"

_ijms, 2023, doi:10.3390/ijms24087137_

Round 1
Reviewer 1 Report
This is a nice review summarising the evidence for inflammation induced by alpha-synuclein as a pathogenic driver in PD and other neurodegenerative conditions. The text is well written, very readable and informative, with a good structure that develops the main messages sequentially. The figures are well designed and informative, complementing the text well.
I have only three minor comments for potential additions, and two are related to potential therapeutic options based on the information provided.
1. Page 3 line 93. If a single missense/point mutation is defined in the a-Syn gene, is there not a possibility for CRISPR/Cas9-mediated therapy? This could be mentioned as a possibity.
2. The second point relates to the therapeutic possibity of modifying gut or oral microbiota as a therapeutic option. The authors stress the importance of characterising microbiota for diagnostic purposes, and although mention of FMT is made (page 8 line 350) I think this could be expanded so it is clearer to the reader what might be attempted - depletion of resident microbiota in oral cavity or gut and recolonisation with a preferred microbiota that will not induce a proinflammatory response.
3. The last point relates to the studies of oral microbiota (page 6 line 263). It is conceivable that patients that succumb to PD might have a more easily-proinflammatory activated innate immune system (e.g. macrophages) - so that with a lower activation threshold they are more prone to induce proinflammatory responses. So major differences in oral microbiota per se might not be the only factor, but how the individual immune system reacts to even the same bacteriuam might be important. Such differences in monocyte reactivities have been described in autoimmune patients, for example. It could be worth making some mention of this possibe factor in PD pathogenesis.
Author Response
Dear Editorial Board,
We appreciate the opportunity to revise and resubmit our manuscript "Revisiting alpha-synuclein pathways to inflammation" (Manuscript ID ijms-2312879).
We thank the editor and reviewers for their comments, all of which have been considered and addressed.
Changes to the manuscript are highlighted in the revised manuscript. Our point-by-point responses to all comments are detailed below. We are happy to consider further revisions and thank you for your continued interest in our research.
REVIEWER 1:
This is a nice review summarising the evidence for inflammation induced by alpha-synuclein as a pathogenic driver in PD and other neurodegenerative conditions. The text is well written, very readable and informative, with a good structure that develops the main messages sequentially. The figures are well designed and informative, complementing the text well.
I have only three minor comments for potential additions, and two are related to potential therapeutic options based on the information provided.
Answer: We appreciate you review and overall remarks. We have considered all three minors commentaries and added an answer individually to each.
- Page 3 line 93. If a single missense/point mutation is defined in the a-Syn gene, is there not a possibility for CRISPR/Cas9-mediated therapy? This could be mentioned as a possibity.
Answer: We appreciate pointing this out. Indeed, in cases of single missense/point mutation of the a-Syn gene, such approaches could be reasonable enough to be thought. We mentioned this possibility and referenced two reports on this regard (PMID: 31773362; and PMID: 34813019) in a newly written phrase that reads as follows: “and have led to consider the CRISPR/Cas9-mediated technology far beyond in this regard [42,43].” (lines 97-98).
- The second point relates to the therapeutic possibity of modifying gut or oral microbiota as a therapeutic option. The authors stress the importance of characterising microbiota for diagnostic purposes, and although mention of FMT is made (page 8 line 350) I think this could be expanded so it is clearer to the reader what might be attempted - depletion of resident microbiota in oral cavity or gut and recolonisation with a preferred microbiota that will not induce a proinflammatory response.
Answer: Thank you very much for your comment. We fully agree with your suggestion and, in fact, we dedicated section 4 “Inflammatory mitigation and α-Syn” of the manuscript to reflect on the importance of characterizing dysbiotic sights, not only for diagnostic purposes but also to suggest possible positive outcomes in patients with synuclein pathology, in treating such infectious conditions. However, current evidence on that regard is still sparse, and future studies clarifying the biological mechanisms behind oral/gut microbiota and synucleinopathies are urgent. Once those mechanisms become clearer, targeting these dysbiotic diseases might reveal novel therapeutic approaches in synucleinopathies.
- The last point relates to the studies of oral microbiota (page 6 line 263). It is conceivable that patients that succumb to PD might have a more easily-proinflammatory activated innate immune system (e.g. macrophages) - so that with a lower activation threshold they are more prone to induce proinflammatory responses. So major differences in oral microbiota per se might not be the only factor, but how the individual immune system reacts to even the same bacteriuam might be important. Such differences in monocyte reactivities have been described in autoimmune patients, for example. It could be worth making some mention of this possibe factor in PD pathogenesis.
Answer: Indeed, the innate immune system is detrimental to the response to a periodontitis-prone biota. We acknowledge the example provided, yet such topics are still under debate. For example, in what macrophage polarization regards, evidence is still contradictory (PMID: 31152604, PMID: 34950140). For those reasons, we decided to take a more careful review and avoid furthering this topic. We added the following sentence as follows: “In addition, immunological different traits and patterns mediated by oral microbiota might interfere with the pro-inflammatory state in PD, with the individual as well as horizontal and vertical inheritance having a conceivable role.” (lines 261-263).
Reviewer 2 Report
The revision entitled "Revisiting alpha-synuclein pathways to inflammation", Of Lyra et al., try to order the most recent articles regarding α-Syn and α-synucleinopathies (from Lewy Bodies aggregations to Parkinson’s Disease).
In this review, they try to summarize the current knowledge on α-Syn mechanistic pathways to inflammation; the eventual role of microbial dysbiosis on α-Syn and exploring the possible influence of inflammatory mitigation. The conclusion of this review tries to clarify the pathophysiological processes underlying α-synucleinopathies, in order to consider the mitigation of existing low-grade chronic inflammatory states, with the aim of start searching for concrete clinical recommendations for this particular condition.
I am very happy for having had the opportunity to read this review that I find easy and quickly reading and after reading this work I have no specific requests for the authors and I consider the work done so far good for publication.

Author Response
Dear Editorial Board,
We appreciate the opportunity to revise and resubmit our manuscript "Revisiting alpha-synuclein pathways to inflammation" (Manuscript ID ijms-2312879).
We thank the editor and reviewers for their comments, all of which have been considered and addressed.
REVIEWER 2:
The revision entitled "Revisiting alpha-synuclein pathways to inflammation", Of Lyra et al., try to order the most recent articles regarding α-Syn and α-synucleinopathies (from Lewy Bodies aggregations to Parkinson’s Disease).
In this review, they try to summarize the current knowledge on α-Syn mechanistic pathways to inflammation; the eventual role of microbial dysbiosis on α-Syn and exploring the possible influence of inflammatory mitigation. The conclusion of this review tries to clarify the pathophysiological processes underlying α-synucleinopathies, in order to consider the mitigation of existing low-grade chronic inflammatory states, with the aim of start searching for concrete clinical recommendations for this particular condition.
I am very happy for having had the opportunity to read this review that I find easy and quickly reading and after reading this work I have no specific requests for the authors and I consider the work done so far good for publication.
Answer: We are satisfied with this reviewer’s overall remark and commentaries.